# Implications of Crop Rotation and Fungicide on *Fusarium* and Mycotoxin Spectra in Manitoba Barley, 2017–2019

**DOI:** 10.3390/toxins14070463

**Published:** 2022-07-06

**Authors:** M. Nazrul Islam, Mitali Banik, Srinivas Sura, James R. Tucker, Xiben Wang

**Affiliations:** 1Agriculture and Agri-Food Canada (AAFC), Morden Research and Development Centre, 101 Route 100, Morden, MB R6M 1Y5, Canada; naz.islam@agr.gc.ca (M.N.I.); mitali.banik@agr.gc.ca (M.B.); srinivas.sura@agr.gc.ca (S.S.); 2Agriculture and Agri-Food Canada (AAFC), Brandon Research and Development Centre, Brandon, MB R7C 1A1, Canada; james.tucker@agr.gc.ca

**Keywords:** *Fusarium* head blight, barley, mycotoxin, chemotype, fungicide and crop rotation

## Abstract

*Fusarium* head blight (FHB) is one of the most important diseases of barley in Manitoba province (western Canada), and other major barley producing regions of the world. Little is known about the *Fusarium* species and mycotoxin spectra associated with FHB of barley in Manitoba. Hence, barley grain samples were collected from 149 commercial fields from 2017 to 2019, along with information on respective cropping history, and analyzed with respect to *Fusarium* species spectra, abundance, chemotype composition, and mycotoxin profiles. *Fusarium poae* was the predominant *Fusarium* species associated with FHB of barley in Manitoba, followed by *F. graminearum*, and *F. sporotrichioides*; *F. equiseti* and *F. avenaceum* were also detected but at low levels. *F. poae* strains with the nivalenol (NIV) chemotype and *F. graminearum* strains with 3-acetyl deoxynivalenol (3-ADON) and 15-acetyl deoxynivalenol (15-ADON) chemotypes were commonly detected in the barley grain samples. Nivalenol (597.7, 219.1, and 412.4 µg kg^−1^) and deoxynivalenol (DON) (264.7, 56.7, and 65.3 µg kg^−1^) were the two most prevalent mycotoxins contaminating Manitoba barley in 2017, 2018 and 2019, respectively. A substantially higher DON content was detected in grain samples from barley fields with cereals as a preceding crop compared to canola and flax. Furthermore, *F. poae* proved less sensitive to four triazole fungicides (metconazole, prothioconazole+tebuconazole, tebuconazole, and prothioconazole) than *F. graminearum*. Findings from this research will assist barley producers with improved understanding of FHB threat levels and optimizing practices for the best management of FHB in barley.

## 1. Introduction

Barley (*Hordeum vulgare* L.) is one of the world’s major cereal crops and known for many health attributes, including being high in soluble fibre, antioxidants, essential vitamins, and minerals that contribute to lowering cholesterol [1]. Globally, barley is harvested at approximately 49.5 million hectares and produces approximately 143 million metric tonnes annually (https://www.fao.org/faostat/en/#data/QCL/visualize, accessed on 6 June 2022). Canada is a major exporter of barley, contributing ~9% of global production. Canadian breweries and livestock operations that rely on barley export over $2 billion in goods to the United States, Japan, South Korea, and Mexico (https://cafta.org/export/barley/, accessed on 6 June 2022). Over 90% of Canadian barley is produced in the western Canadian provinces of Manitoba, Saskatchewan, and Alberta.

Since the 1990s, outbreaks of *Fusarium* head blight (FHB) on barley have been frequently reported in western Canadian provinces and this disease has been recognized as the most important disease of barley in these regions [2,3,4]. The main concern with respect to FHB on barley is the potential accumulation of mycotoxins in harvested grains, which can be toxic to humans and animals [5,6,7,8,9]. In addition, these mycotoxins can negatively impact functional parameters related to barley malting and brewing quality (i.e., over foaming/gushing of beer) [10,11].

Fusarium head blight of barley is caused by a complex of toxigenic *Fusarium* species, including *F. graminearum* (W.G. Smith) Sacc., *F. poae* (Peck) Wollenw., *F. culmorum* (WG Smith) Sacc., *F. avenaceum* (Fr.) Sacc., *F. sporotrichioides* Scherb., *F. equiseti* (Corda) Sacc., *F. cerealis*, and *F. verticilloides* [12]. In North America, *F. graminearum* has been regarded as the primary causal agent of FHB on small grain cereals, especially on wheat. *Fusarium graminearum* is a potent producer of zearalenone (ZEA) and type B trichothecenes, mainly including deoxynivalenol (DON) and nivalenol (NIV). Deoxynivalenol is the most frequently detected trichothecene mycotoxin in cereal grains. The Canadian Food Inspection Agency (CFIA) has set a maximum tolerated level of DON in cereal grains, including barley, at 2 mg kg^−1^ for human consumption and 1 mg kg^−1^ for feed (https://inspection.canada.ca/animal-health/livestock-feeds/regulatory-guidance/rg-8, accessed on 6 June 2022). DON is also highly regulated by the barley brewing and malting industry due to public safety concerns, where exceeding limits as low as 0.5 mg kg^−1^ may result in rejected sale [13]. The European Commission has similar regulations with maximum limits for DON, at 1.25 mg kg^−1^ for unprocessed barley grains and 0.5 mg kg^−1^ for malting barley [6].

*Fusarium graminearum* strains can be categorized into different chemotypes based on their toxigenic profiles, including NIV, 15-acetyl deoxynivalenol (15-ADON), and 3-acetyl deoxynivalenol (3-ADON) [14]. Before the 1990s, *F. graminearum* strains with the 15-ADON chemotype were predominant in North America [15]. However, there has been a gradual shift towards 3-ADON chemotype strains replacing the 15-ADON chemotype in Canada [16]. *Fusarium graminearum* strains capable of producing a novel type A trichothecene called NX-2 have been detected in the northern USA and eastern Canada [17]. *Fusarium graminearum* strains with the NIV chemotype are common in Europe, Japan, Australia, and southern parts of the United States [18]. However, this group of *F. graminearum* strains has not been detected on wheat and barley in western Canada up to the current date [2]. Therefore, information on the extent of NIV contamination in Manitoba barley is limited. In 2015, *F. cerealis* strains with the NIV chemotype were isolated from winter wheat fields in Carman, Manitoba [19]. The discovery of NIV-producing *Fusarium* species in Manitoba has raised concerns about changing mycotoxin profiles and the importance of testing for NIV in cereal grains.

In western Canada, FHB of barley has been primarily associated with *F. graminearum*. Other *Fusarium* species, including *F. poae*, *F. sporotrichioides*, and *F. avenaceum*, have also been isolated from infected barley grains. Moreover, an upward trend in the isolation frequency of *F. poae* in barley has been noted in recent surveys conducted in Canadian provinces [20,21]. Similarly, *F. poae* has become the predominant *Fusarium* pathogen of barley in the United Kingdom and Argentina, replacing *F. graminearum* [22,23,24].

*Fusarium poae* produces both trichothecene and non-trichothecene mycotoxins, including DON, NIV, beauvericin (BEA), enniatins (ENNs), diacetoxyscirpenol (DAS), HT-2 toxin, T-2 toxin, and fusarenone-x (FX) [25,26]. In Canada, the maximum tolerated levels of HT-2 and T-2 toxins for feed are 0.1 and 1 mg kg^−1^, respectively. The recommended tolerance level for DAS in poultry feed is 1 mg kg^−1^ (https://inspection.canada.ca/animal-health/livestock-feeds/regulatory-guidance/rg-8, accessed on 6 June 2022). *Fusarium poae* is thought to be a weak pathogen on wheat and barley compared to *F. graminearum*, as *F. poae* may infect cereal heads but only grow superficially and thrive primarily at the infection site [27,28]. However, the increase in *F. poae* infection in different cereal crops, especially barley and oat, has raised questions about the importance of this pathogen in the FHB disease complex.

To better understand the FHB species spectra and associated mycotoxins affecting Manitoba barley, analysis was carried out on barley grain samples collected from producers’ fields in three consecutive years (2017, 2018, and 2019), from five crop districts of Manitoba. This study investigates four main objectives: (i) *Fusarium* species spectra and chemotypes associated with FHB of barley; (ii) profile of naturally occurring *Fusarium* mycotoxins in barley grains and their main fungal source; (iii) impact of commonly used fungicides on the mycelial growth of *F. poae* and *F. graminearum*, and (iv) influence of crop rotation on DON and NIV levels in barley grains.

## 2. Results

### 2.1. Fusarium Species Spectra and Chemotype Composition

PCR-based assays were used to investigate *Fusarium* species spectra associated with FHB of barley in Manitoba (Figure 1A). A total of five *Fusarium* species were detected from 2017 to 2019. *F. poae* was the most common, detected in 66%, 62%, and 55% of barley samples in 2017, 2018 and 2019, respectively. *F. graminearum* was the next most common, detected in 40 to 57% of barley samples from 2017 to 2019. *F. sporotrichioides* was detected in 43% of barley samples in 2017, but at a much lower rate of 7% and 9% of samples in 2018 and 2019, respectively. Infections caused by *F. avenaceum* and *F. equiseti* were also found but at much lower frequencies compared to those attributed to *F. poae*, *F. graminearum*, and *F. sporotrichioides*.

The chemotype composition of *Fusarium* pathogens associated with FHB of barley in Manitoba was also evaluated (Figure 1B). The percentage of barley samples infected with *F. graminearum* 3-ADON strains was slightly higher (65%, 50%, and 45% of samples) than for 15-ADON strains (58%, 38%, and 40% of samples) from 2017 to 2019, respectively. *Fusarium graminearum* strains with the NIV chemotype were not detected. In comparison, *F. poae* strains with the NIV chemotype were detected in 100% of barley samples from 2017 and 2019 and in 82% of samples from 2018.

### 2.2. Abundance of Fusarium DNA

Quantitative PCR was used to analyse the abundance of *Fusarium* gDNA in barley grains (Figure 2). *Fusarium poae* was the most abundant *Fusarium* species in 2017 and 2018. Mean concentrations of *F. poae* gDNA in 2017 and 2018 samples were 0.74 and 0.16 pg *Fusarium* gDNA ng^−1^ total gDNA, significantly higher than mean concentrations of *F. graminearum* gDNA at 0.20 and 0.06 pg *Fusarium* gDNA ng^−1^ total gDNA, respectively.

In 2019 barley samples, the mean concentrations of *F. poae* and *F. graminearum* gDNA were similar at 0.08 and 0.09 pg *Fusarium* gDNA ng^−1^ total gDNA, respectively. The abundance of *F. sporotrichioides* gDNA in barley grains was much lower and varied from 0.003 to 0.01 pg *Fusarium* gDNA ng^−1^ total gDNA from 2017 to 2019. Due to low Ct values (below 39) in most samples, data for *F. avenaceum* and *F. equiseti* were excluded from the analysis.

### 2.3. Fusarium Mycotoxins

Naturally occurring mycotoxins in barley grains were analysed using UHPLC-HRMS (Table 1). The two most common mycotoxins found in barley grain samples in Manitoba, were NIV and DON. From 2017 to 2019, NIV was detected in 95.5%, 91.1%, and 71.2% of barley samples (2017, 2018, and 2019, respectively) and at concentrations exceeding 500 µg kg^−1^ in 29.5% (2017), 16.7% (2018) and 22.4% (2019) of the samples. This was considerably higher than the percentage of barley samples contaminated with DON and exceeding the concentration of 500 µg kg^−1^ (6.8%, 4.5%, and 3.4% of samples from 2017, 2018, and 2019, respectively). The mean DON concentration in barley was 264.7, 56.7, and 65.3 µg kg^−1^ in 2017, 2018, and 2019 samples, respectively, with these values significantly lower than for NIV (597.7, 219.1, and 412.4 µg kg^−1^ in 2017, 2018, and 2019, respectively). Beauvericin was detected in 47.5 to 74.7% of samples with mean concentrations ranging between 9.3 and 14.5 µg kg^−1^. The maximum concentration of BEA observed for any sample was 211.4 µg kg^−1^, which was much lower than for NIV (4136.6 µg kg^−1^) or DON (8921.6 µg kg^−1^). Other mycotoxins including T-2, HT-2, CULM, DAS, and ENNs (A, A1, B, and B1) were also detected in the barley grain samples in all three years, however, at low average concentrations, ranged between 0.3–2.8 µg kg^−1^ for T-2, 0.3–9.9 µg kg^−1^ for HT-2, 2.5–15.8 µg kg^−1^ for CULM, 0.3–0.9 µg kg^−1^ for DAS, and 14.7–27.4 µg kg^−1^ for ENNs between 2017 and 2019.

### 2.4. PCA of the Concentrations of Fusarium gDNA and Mycotoxins in Barley Grains

The results from the quantification of *Fusarium* gDNA and mycotoxins in barley grain samples were subjected to PCA (Figure 3). The first two dimensions account for 45.74% of the variability: component one on the x-axis describes 25.11% of the variability and component two on the y-axis represents an additional 20.63%. A high correlation was found between the concentrations of *F. graminearum* gDNA and DON/CULM as they clustered in the same quadrant. Concentrations of *F. poae* gDNA and NIV/BEA /DAS also clustered in the same quadrant. Furthermore, *F. sporotrichioides* gDNA and T-2/HT-2 concentrations also showed a high correlation. No clear correlation was established between any of the *Fusarium* species and ZEA/ENNs (Figure 3).

### 2.5. Sensitivity of F. poae and F. graminearum Strains to Triazole Fungicides

The sensitivity of *F. poae* and *F. graminearum* to metconazole, prothioconazole+tebuconazole, tebuconazole, and prothioconazole was analysed using PDA plate-based assays supplemented with two fungicide concentrations (0.01 and 0.1 mg of active ingredient L^−1^) (Table 2). Compared to Proline and Folicur, Caramba and Prosaro displayed a higher inhibitory effect against the mycelial growth of *F. poae* and *F. graminearum*. Interestingly, *F. poae* strains showed lower sensitivity to all four fungicides than *F. graminearum* strains at both concentrations, except for Caramba at 0.1 mg active ingredient L^−1^ and Folicur at 0.01 active ingredient L^−1^ (Table 2).

### 2.6. Effects of Preceding Crop and Year on Fusarium Mycotoxins in Barley Grains

The effect of preceding crop and year on DON and NIV concentrations in barley samples was investigated using the GLIMMIX procedure (Table 3). Overall, previous crop alone and interactions between year and previous crop significantly (*p* ≤ 0.05) affected DON concentrations in barley samples from Manitoba. Cereal–barley rotations resulted in the highest DON concentrations in barley grains compared to canola–barley or flax–barley rotations. However, NIV concentrations were similar in barley samples from different preceding crop’s combinations (cereal–barley, canola–barley or flax–barley).

## 3. Discussion

The current study investigated populations of *Fusarium* pathogens and mycotoxin profiles in grains from naturally infected commercial barley fields from five crop districts in Manitoba. The results indicate that the *Fusarium* species composition associated with FHB of barley in Manitoba is diverse and dominated by several toxigenic *Fusarium* species. *F. poae* was the predominant species, followed by *F. graminearum* and *F. sporotrichioides*. This pattern agrees with the results of a 13-year barley FHB survey (2005 to 2017) in eastern Canada, which showed *F. poae* and *F. graminearum* were the predominant *Fusarium* pathogens in barley at that time, representing 42% and 40%, respectively, of the total *Fusarium* population in the region [29]. *F. poae* has also been reported as the predominant *Fusarium* pathogen associated with FHB of barley in several South American and European countries [22,30].

The *Fusarium* species spectra associated with FHB of barley appear be different than that affecting wheat. *Fusarium graminearum* is typically considered the most prevalent and aggressive FHB pathogen of wheat in the temperate and warmer regions of the United States and Canada [31,32], while in cooler regions of Europe *F. graminearum*, *F. culmorum* and *F. avenaceum* are predominant [33]. Therefore, research on *Fusarium* pathogens associated with FHB of barley in these regions has primarily focused on *F. graminearum*. Our findings, together with others, show *Fusarium* species other than *F. graminearum*, such as *F. poae* and *F. sporotrichioides*, play a significant role in the FHB disease complex on barley. This result is of particular importance and suggests more focus should be directed towards understanding the impact of *Fusarium* species previously considered less aggressive but still economically important due to their association with FHB and mycotoxin accumulation and subsequent cumulative negative impacts on the grain yield and quality.

Compared to other *Fusarium* species involved in FHB, *F. poae* causes the least reduction in seed germination [34] and its infection on the cereal head is mainly asymptomatic [35]. Therefore, *F. poae* had been considered a weak pathogen on cereals. However, recent studies show *F. poae* has become omnipresent in Europe, South America, and North America [36,37,38], which raises concerns about the importance of this pathogen in the FHB disease complex.

Climatic conditions, specifically temperature and humidity, can significantly impact the distribution and predominance of *Fusarium* species. An extended period of wet conditions during plant anthesis is well documented as crucial for the infection of *F. graminearum* [39]. On the contrary, humidity duration has no significant effect on the infection of *F. poae* and its mycotoxin production [40]. The optimal temperature of *F. poae* growth is between 25 and 30 °C, similar to that of *F. sporotrichioides* but slightly higher than *F. graminearum* (20–25 °C) [41]. *F. poae* may proliferate well in hot and dry conditions rather than the wet and cool conditions preferred by *F. graminearum*. Manitoba experienced relatively warm and dry conditions during growing seasons and barley anthesis between 2017 and 2019. The predominance of *F. poae* over other *Fusarium* species, especially *F. graminearum*, can be partially explained by the occurrence of thermo-hygrometric conditions that do not favour *F. graminearum*, allowing for a shift in the population dynamics towards *F. poae*.

Concentrations of NIV and *F. poae* gDNA were highly correlated in barley grain samples from Manitoba. These results agree with the findings of Yli-Mattila et al. [42], where a similar high correlation in barley samples from Finland and Russia was observed. Islam et al. [37] reported correlations between NIV content and the concentration of *F. poae* DNA in oat samples from Manitoba. These results suggest *F. poae* is a potent NIV producer in barley. In addition, BEA and DAS are also highly correlated with *F. poae* infection. However, these two mycotoxins are only produced at low levels.

In this study, 16.7% to 29.5% of barley grain samples were contaminated with NIV at concentrations higher than 500 µg kg^−1^. The maximum concentration of NIV was 4136.6, 3252.1, and 2245.4 µg kg^−1^ in 2017, 2018, and 2019, respectively. These values are in agreement with the results of Tucker et al. [43], who detected a high level of NIV contamination in grain samples from various mock-inoculated (water) barley genotypes in an FHB screening disease nursery using *F. graminearum* as the inoculum in Brandon, Manitoba. The extent of NIV contamination found in barley in this study is higher than that reported in the oat crop. Specifically, a survey of *Fusarium* mycotoxins in oat grains from Manitoba, collected between 2016 to 2018, found maximum concentrations of NIV ranged between 581 to 865 µg kg^−1^ [37]. To date, NIV contamination in wheat has been rarely reported in Western Canada and is mainly due to the predominance of *F. graminearum* on wheat and the absence of *F. graminearum* strains with the NIV chemotype in the region [31].

The significant level of NIV contamination in naturally infected barley grain is a cause of concern for food safety and security. NIV is more toxic to animals than DON due to the presence of a hydroxyl group, instead of hydrogen, at the C-4 position. The presence of a hydroxyl group in NIV increases its toxicity to animals by ten-fold compared to DON [6]. The European Food Safety Administration established a lower tolerated daily intake of 0.7 µg kg^−^^1^ body weight for NIV compared to 1 µg kg^−^^1^ for DON. Tolerance limits for NIV in food cereals have not been identified yet in Canada [44] and currently, NIV concentrations in barley grains are neither routinely determined nor legislatively regulated in Canada [19]. The discovery of NIV contamination associated with *F. poae* in barley shows the importance of routine testing for NIV in naturally affected barley grain samples and the need for more extensive research on *F. poae*. These results also have practical implications, such as the need to develop FHB-resistant barley varieties and conduct research and monitor beyond the current sole focus on *F. graminearum* and DON [2,4,45].

In this study, *F. graminearum* and DON were detected at lower levels than *F. poae* and NIV in barley grains. These observations were expected and align with the prevailing weather conditions in Manitoba i.e., relatively dry conditions in June and July of 2017, 2018, and 2019 that are not very conducive to infection by *F. graminearum* resulting in low DON concentrations. However, the maximum concentrations of DON in barley samples reached 8921.6, 1488.8, and 2051.1 µg kg^−1^ in these three respective years. This indicates *F. graminearum* can still pose a significant threat to growers when regional microenvironment conditions favour it as a causal agent.

Both *F. graminearum* 3-ADON and 15-ADON chemotype strains were detected in barley samples from Manitoba, with 3-ADON strains being detected at a higher frequency. Similarly, a higher percentage of *F. graminearum* strains with the 3-ADON chemotype were found in wheat samples from Western Canada [46,47]. Before 1994, *F. graminearum* strains with the 15-ADON chemotype were the principal pathogen causing FHB in wheat in North America. However, a shift to *F. graminearum* strains with the 3-ADON chemotype was documented between 1998 and 2004, attributed to changes in environmental conditions, host distribution, and various agricultural practices [46]. *F. graminearum* strains with the NIV chemotype were not detected in this study. This result agrees with previous studies on the *F. graminearum* population in Western Canada that suggest the lack of this group of *F. graminearum* strains in the region [31,37].

*Fusarium sporotrichioides* infection was moderate in 2017 and relatively low in 2018 and 2019. Concentrations of HT-2 and T-2 detected in barley samples were comparable to results from previous studies of naturally infected barley samples. In a study, investigating the natural contamination in barley in Italy, Lattanzio et al. [48] reported maximum concentrations of HT-2/T-2 up to 220 and 198 µg kg^−1^ in cleaned barley and malt samples, respectively, collected in 2013. Kis et al. [49] found 12.2 to 52.1 µg kg^−1^ of HT-2/T-2 in barley samples from three Croatian regions. In Western Canadian provinces, the problem of *F. sporotrichioides* and HT-2/T-2 contamination in barley varies from year to year. In the summer of 1993 with relatively high rainfall, the level of *F. sporotrichioides* reached 30% in barley in Manitoba [50]. In a field study at an experimental farm in southern Manitoba, with barley varieties inoculated with *F. sporotrichioides*, HT-2/T-2 concentrations reached 670 to 3720 µg kg^−1^ [4]. These results suggest a significant level of HT-2/T-2 contamination could occur when environmental conditions are conducive to *F. sporotrichioides*. Due to barley being a major component of many feed formulations, livestock toxicity of HT-2/T-2 might arise in years with high *F. sporotrichioides* infection. Therefore, feed mill operators should consider chemical assays to ensure their feed components contain minimal amounts of HT-2/T-2.

To evaluate the potential effect of agrochemicals on the *Fusarium* population, we analysed the sensitivity of *F. poae* and *F. graminearum* to four triazole fungicides commonly used in Manitoba to manage FHB. The results showed *F. poae* has lower sensitivity to these fungicides than *F. graminearum*. Similarly, Tini et al. [51] show *F. poae* strains from Italy are more resistant to fungicides containing a similar group of active ingredients than *F. graminearum* and *F. culmorum*. In addition, Audenaert et al. [52] report the treatment of wheat fields with azole fungicides caused the *Fusarium* population to shift from one dominated by *F. culmorum* and *F. graminearum* to one dominated by *F. poae*. These results suggest the pressure exerted by the repeated use of fungicides may impact *Fusarium* population dynamics, leading to the shift in the pathogen population towards those more resistant to these fungicides.

In this study, cereal–barley rotations resulted in higher DON levels in barley grains than canola–barley or flax–barley rotations. This result agrees with findings by Schöneberg et al. [53], in which the cultivation of corn before barley caused a higher *F. graminearum* incidence and DON content than a canola–barley rotation. Pageau et al. [54] also report the DON content in barley grains was higher when cereals, rather than dry pea, were seeded in the rotation. Interestingly, no significant impact of previous crops on NIV concentration in barley was observed. Similarly, Vogelgsang et al. [38] show cropping history had no impact on the incidence of *F. poae* and NIV content in wheat. Differences in the effects of the previous crop on the concentration of DON and NIV in barley suggest *F. poae* and *F. graminearum* adapt differently to crop residues. Crop rotations that are effective in managing *F. graminearum*/DON might not be suitable for limiting *F. poae* infection and NIV accumulation in barley. We speculate the lesser impact of cropping rotation on the abundance of *F. poae* and associated mycotoxins could be due to the broad host range of *F. poae*, which includes small grain cereals, soybean, sunflower, and various broad-leaf grasses [55,56,57].

The multi-species nature of FHB on barley also raises the question about the impact of interactions between different *Fusarium* species during infection and consequent mycotoxin production in barley. Tan et al. [36] show pre-inoculation of wheat ears with *F. poae* before *F. graminearum* led to a disappearance of FHB symptoms and reduced mycotoxin levels compared to a single *F. graminearum* infection, while *F. poae* exhibited increased growth. Moreover, *F. poae* can asymptomatically induce salicylic acid and jasmonic acid related defences, which could hamper a later infection by *F. graminearum* [58]. Additionally, Walkowiak et al. [59] report co-inoculation with *F. graminearum* 3-ADON and 15-ADON strains results in a reduction in infection and toxin accumulation in planta. Evidence also suggests the co-occurrence of DON and NIV might synergistically affect their cytotoxicity [60].

## 4. Conclusions

The current study investigated barley grain collected in Manitoba for *Fusarium* species spectra and naturally occurring mycotoxins. We demonstrate that *Fusarium* spectra associated with FHB of barley in Manitoba are diverse and mainly included *F. poae, F. graminearum*, and *F. sporotrichioides*. NIV and DON are the two most common mycotoxins detected in barley grains in Manitoba. Results also demonstrated that disease management practices, such as fungicide and crop rotation, and weather conditions can impact the *Fusarium* species spectra and their respective mycotoxins in barley. Further monitoring of FHB in barley is imperative to confirming our findings on a greater temporal and spatial scale and to observe potential changes in the *Fusarium* species spectra and their respective mycotoxins. Our results also address the need for more research on the pathology of *F. poae* and *F. sporotrichioides* on barley.

## 5. Materials and Methods

### 5.1. Barley Sample Collection

Manitoba is located in the centre of the North American continent and its climate mainly falls into the humid continental climate zone. In general, the anthesis period of barley in Manitoba is between 1 to 15 July. In July, the mean temperatures of Manitoba were 19.8 °C, 20.0 °C, and 19.6 °C in 2017, 2018, and 2019, respectively. The mean precipitations of the province in July were 47.7 cm, 48.4 cm and 59.0 cm in 2017, 2018, and 2019, respectively (Appendix A). A total of 149 barley fields (44,47, and 58 in 2017, 2018, and 2019, respectively) from five crop districts of Manitoba (Central, Southwest, Eastern, Northwest, and Interlake; Latitude N 49°1′35.311″N 51°13′41.345″; Longitude W 97°8′50.631″W 101°21′6.934″; Appendix A) were surveyed for FHB between July 18 and August 5 of each year. The fields were surveyed between the late milk growth stage (BBCH 77) and full maturity (BBCH 89). Barley spikes (*n* = 40 to 60) were randomly collected from five locations in each field and stored in paper envelopes for subsequent laboratory analysis. Information on the preceding crops (canola, flax or cereal) was also collected for each field/sample location.

### 5.2. DNA Extraction and PCR Analysis of Fusarium Genomic DNA

A subsample of 20 g of barley grains per field was ground to flour using a Retsch grinder (Retsch ZM 200, Scientific Inc. Newtown, PA, USA). Total genomic DNA (gDNA) was extracted from 1 g of flour using a QIAGEN DNeasy Mini Kit (QIAGEN Mississauga, ON, Canada) following the manufacturer’s procedure.

Multiplex PCR was performed to determine *Fusarium* species spectra (*F. poae*, *F. graminearum*, *F. sporotrichioides*, *F. avenaceum*, and *F. equiseti*) and chemotype composition (Appendix A). PCR assays were carried out in a BioRad C1000 thermal cycler (BioRad, Mississauga, ON, Canada) using the parameters described in Islam et al. [37]. PCR amplicons were separated on 2% agarose gels in 1 × TAE buffer and stained with GelRed (Biotium, Mississauga, ON, Canada). Gel images were scanned into Gel Doc™ EZ Imager (BioRad, Mississauga, ON, Canada).

Quantitative PCR was used to determine the abundance of *Fusarium* gDNA in barley using a CFX96™ Real-Time PCR Detector System (BioRad, Mississauga, ON, Canada). All standards and the negative control (double-distilled water) were run in triplicate. The PCR reaction was carried out in a total volume of 20 µL, consisting of 10 µL of Fast EvaGreen^®^ PCR Master Mix (BioRad, Mississauga, ON, Canada), 1 µL of each primer, 6 µL of double-distilled water, and 2 µL of template DNA with a 37-cycle threshold (Ct) detection limit. PCR conditions were as follows: initial preheating at 98 °C for 2 min; 40 cycles of 95 °C for 15 s and 62 °C for 1 min; and dissociation curve analysis at 60 to 95 °C. Three technical replicates were performed for each sample.

### 5.3. Detection and Quantification of Fusarium Mycotoxins

Barley grain samples were analysed for *Fusarium* mycotoxins, including nivalenol (NIV), deoxynivalenol (DON), 3-acetyldeoxynivalenol (3-ADON), 3-acetyldeoxynivalenol (15-ADON), deoxynivalenol-3-glucoside (DON3G), diacetoxyscirpenol (DAS), beauvericin (BEA), HT-2, T-2, enniatins (ENN A, A1, B, and B1), zearalenone (ZEA), culmorin (CULM), and fumonisin-B1 (FB1) in-house at Morden Research and Development Centre, Agriculture and Agri-Food Canada. Ground barley samples (1 g) were extracted for mycotoxins using a solvent mixture (10 mL; 75% acetonitrile: 10% methanol: 15% water) in a 10-mL flat-bottomed tube. The sample-solvent mixture was thoroughly mixed by inversion to wet the flour sample completely, followed by sonication (30 min). The tubes were then loaded onto a rotary shaker and extracted for 90 min at 40 rpm. The sample-solvent mixture was centrifuged (4000× *g* for 30 min) to separate the solvent from the ground material and the supernatant was filtered through a 0.2-µm-nylon syringe filter (ThermoFisher, Mississauga, ON, Canada) into a clean 10-mL flat-bottomed tube. An aliquot of the filtered extract was dried under a gentle stream of nitrogen gas using a sample evaporator (RapidVap Labconco, Kansas City, MO, USA) at 100% speed for 90 min at 70 °C. Each dried-sample extract was dissolved and re-suspended in 1 mL of solvent mixture (50:50, water + 0.1% formic acid + 5 mM ammonium formate: methanol + 0.1% formic acid + 5 mM ammonium formate) by vortexing and brief sonication. The extract was transferred into an amber liquid chromatography vial before analysis using ultra-high performance liquid chromatography (UHPLC; Vanquish, Thermo Fisher Scientific, Mississauga, ON, Canada) coupled with high-resolution mass spectrometer (HRMS; Orbitrap ID-X Tribrid Mass Spectrometer, Thermo Fisher Scientific Inc., Mississauga, ON, Canada).

Separation of various mycotoxins was achieved using a reverse-phase C-18 core-shell silica column (particle size 1.7 µm, 100 × 2.1 mm, Kinetex, Phenomenex, CA, USA) held at 35 °C. Gradient elution was achieved with 100% water (mobile phase A) and 100% methanol (mobile phase B), with both phases containing 0.1% formic acid and 5 mM ammonium formate, at a flow rate of 0.2 mL min^−1^ for 20 min. Retention times and recoveries for various mycotoxins are identified in Appendix A. Heated-electrospray ionization (H-ESI) was used to achieve a steady state electrospray from the chromatographic separation column. Mass spectra for each sample was obtained in positive mode with an obitrap mass resolution of 120,000 for precursor ion mass-to-charge (m/z) values with a mass tolerance of ±5 ppm; product ions mass spectra were acquired at with an orbitrap resolution of 30,000. Precursor and product ion m/z values are noted in Appendix A. Fragmentation of precursor ions was achieved using a higher energy collision-induced dissociation (HCD) with stepped collision energies (15%, 20%, and 25%). The precursor ion mass was used for analyte quantification while product ions were used for analyte confirmation. Spiked reference material was simultaneously extracted with each batch of samples for quality control.

### 5.4. Sensitivity of F. poae and F. graminearum to Triazole Fungicides

The sensitivity of *F. poae* and *F. graminearum* to triazole fungicides, including Caramba^®^ (metconazole, BASF Canada), Prosaro^®^ (prothioconazole + tebuconazole, BAYER Canada), Folicur^®^ (tebuconazole, BAYER Canada), and Proline^®^ (prothioconazole, BAYER Canada), was analysed using a potato-dextrose-agar (PDA)-based petri plate assay, supplemented with different concentrations of fungicide active ingredients. Fifteen *F. poae* and 15 *F. graminearum* strains were included in the analysis. These *Fusarium* strains were randomly selected from the *Fusarium* strains isolated from barley samples collected from 2017 to 2019. The in vitro assay was carried out in triplicate at two doses (0.01 and 0.10 mg of active ingredient (ai)/L PDA). The incubation protocol of the plates was as described by Spolti et al. [15]. The petri plates were incubated at 25 °C for 3 days, and then the mycelial growth of *Fusarium* strains measured. Mycelial growth inhibition (colony diameters on each plate) was determined according to:$$\%\;inhibition=\frac{radial\;growth\;of\;untreated\;control\;plate-radial\;growth\;of\;treated\;plate}{radial\;growth\;of\;untreated\;control\;plate}\times 100$$

### 5.5. Statistical Analysis

For statistical analyses, the homogeneity of *Fusarium* species abundance (qPCR) and mycotoxin concentrations (UHPLC-HRMS) from the barley fields/samples (*n* = 149 over 3 years) was verified using the Kolmogorov–Smirnov test of normality, which measures the divergence of the field sample distribution. The skewness and kurtosis were determined and plotted for each year of survey data to determine whether the data distribution was normal. When necessary, the data were transformed (log, square root, arcsine) or dataset outliers (extreme low and high values) were removed. NIV and DON concentrations were described with box and whisker plots using Microsoft Excel 2016, showing the distribution of mycotoxin concentrations in terms of the mean, median, minimum, maximum, and outliers. The least significant difference (LSD) test was performed to compare mean concentrations of *Fusarium* species gDNA. Student’s *t*-test was used to compare the pair means of mycotoxins (DON and NIV) within each sampling year.

Principal component analysis (PCA) was performed to analyse correlations between the concentrations of *Fusarium* species (*F. poae*, *F. graminearum*, *F. sporotrichioides* from *n* = 149 samples) DNA and nine *Fusarium* mycotoxins (DON, NIV, BEA, HT2, T2, CULM, DAS, ENNs, and ZEA) using SAS v.9.4 (SAS Institute Inc. Cary, NC, USA).

To determine the impact of crop rotation and survey year on the mean concentration of DON and NIV in barley grains, the GLIMMIX procedure in SAS v.9.4 was used to estimate the fixed and random effects of previous crop (PC), year (Y), and PC × Y interaction. Multiple comparisons of means were made using the Tukey–Kramer test (alpha = 0.05). A one-way ANOVA was performed to test the significance of previous crops on DON and NIV concentrations using data from all three years. SAS v.9.4 was also used to identify the significance of fungicides and doses in controlling the growth of *Fusarium* strains. Differences in the inhibition of the mycelial growth of *F. graminearum* and *F. poae* against triazole fungicides were analysed using Student’s *t*-test.

## Figures and Tables

**Figure 1 toxins-14-00463-f001:**
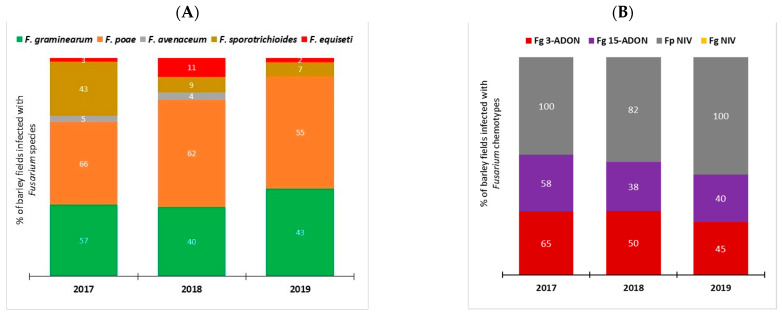
PCR-based identification of *Fusarium* species complex and chemotypes associated with Fusarium head blight on barley in Manitoba (2017–2019). (**A**) Stacked bar plot of *Fusarium* species DNA detected in commercial barley fields in Manitoba; (**B**) Distribution of *Fusarium* chemotypes detected in barley fields in Manitoba. Data from 149 samples or fields (2017, *n* = 44; 2018, *n* = 47; 2019, *n* = 58) were used for this analysis.

**Figure 2 toxins-14-00463-f002:**
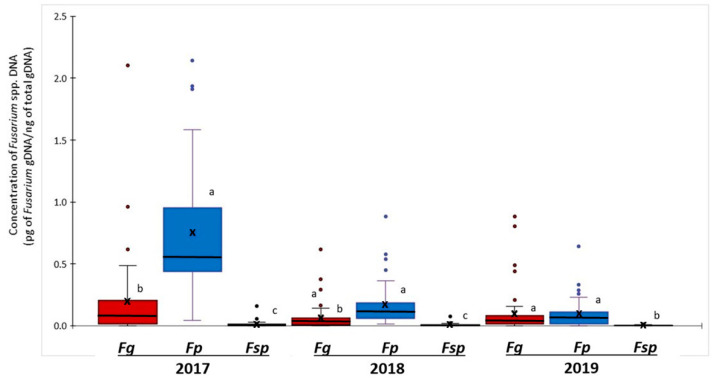
Quantitative PCR analysis of *F. graminearum (Fg)*, *F. poae (Fp)*, and *F. sporotrichioides (Fsp)* DNA in barley grains from Manitoba (2017–2019). Mean and median concentrations are respectively indicated by crosses and horizontal lines in the boxes. Vertical lines outside the boxes are the standard deviation (SD). Outliers are indicated by the coloured circles outside the SD range. Means followed by the same letter within the same year are not significantly different at *p* = 0.05.

**Figure 3 toxins-14-00463-f003:**
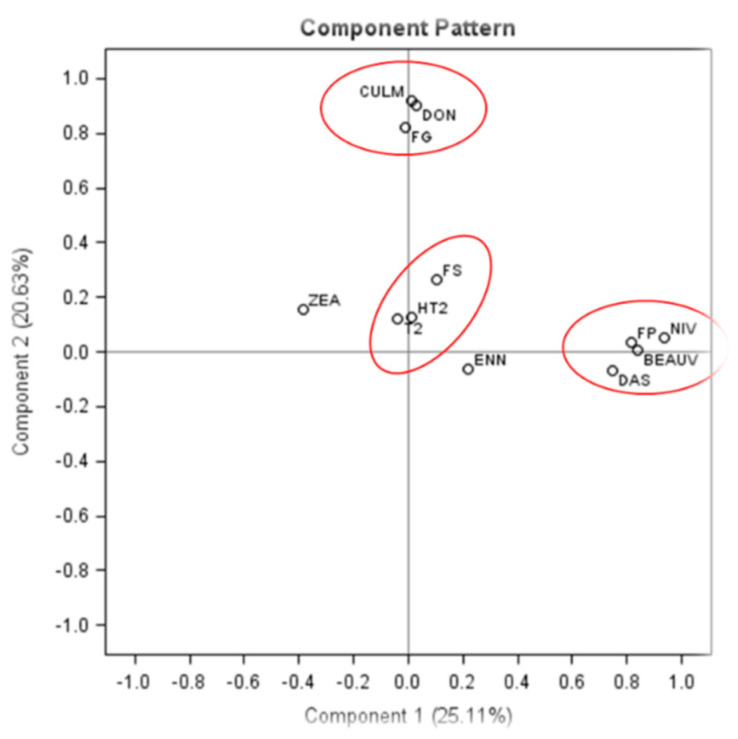
Principal component analysis (PCA) showing the major mycotoxins and *Fusarium* spp. associated with barley grains. The statistical significance of relationships between species and toxins was determined by SAS corelation coefficient and probability analysis (*F. poae*/FP: NIV, *p* < 0.0001, *r* = 0.88, BEAUV, *p* < 0.0001, *r* = 0.47, DAS, *p* < 0.0001, *r* = 0.39; *F. graminearum*/FG: DON, *p* < 0.0001, *r* = 0.63, CULM, *p* < 0.0001, *r* = 0.65; *F. sporotrichioides*/FS: HT2, *p* < 0.0001, *r* = 0.37, T2, *p* < 0.001, *r* = 0.24, ZEA, *p* = 0.111, *r* = −0.13, ENN, *p* = 0.851, *r* = −0.015).

**Table 1 toxins-14-00463-t001:** *Fusarium* Mycotoxins in Barley Grains from Manitoba (2017–2019).

Year	2017				2018				2019			
Samples	44				47				58			
Mycotoxins	Mean (µg kg^−1^)	Maximum (µg kg^−1^)	% of Samples above LOD	% of Samples above 500 (µg kg^−1^)	Mean (µg kg^−1^)	Maximum (µg kg^−1^)	% of Samples above LOD	% of samples above 500 (µg kg^−1^)	Mean (µg kg^−1^)	Maximum (µg kg^−1^)	% of Samples above LOD	% of Samples above 500 (µg kg^−1^)
**NIV**	597.7	4136.6	95.5	29.5	219.1	3252.1	91.1	16.7	412.4	2245.4	71.2	22.4
**DON**	264.7	8921.6	54.5	6.8	56.7	1488.8	31.1	4.5	65.3	2051.1	25.4	3.4
**DON-3G**	87.6	3117.4	53.5	2.3	3.9	20.5	20.5	-	18.9	443.3	23.7	-
**3-ADON**	2.8	79.9	13.9	-	3.2	98.8	6.8	-	2.3	87.2	3	-
**15-ADON**	0.9	32.8	2.3	-	1.1	44.7	2.3	-	2.1	111.8	1.7	-
**BEAUV**	9.3	86.8	74.7	-	10.4	211.4	59.1	-	14.5	122.5	47.5	-
**HT-2**	9.9	120.7	11.4	-	0.3	51.5	4.3	-	5.1	278.1	6.7	-
**T-2**	2.8	108.8	6.8	-	0.3	45.5	4.3	-	2.7	137.1	6.7	-
**ENNs**	21.7	371.8	21.5	-	14.7	266.1	18.2	-	27.4	1046.5	5.1	3.4
**CULM**	10.7	298.3	27.3	-	2.5	65.1	11.4	-	15.8	385.2	27.1	-
**DAS**	0.9	25.7	4.5	-	0.3	12.7	4.3	-	0.3	10.7	3.4	-
**ZEA**	2.4	93.5	9.1	-	0.3	15.2	6.3	-	0.3	2.5	3.4	-

Notes: Limit of quantitation (LOQ) and limit of detection (LOD) were 2.5 and 1 µg kg^−1^, respectively. For calculation of the mean, mycotoxin concentrations below the respective LOQ or LOD were calculated as LOQ/2 or LOD/2, respectively. ENNs is the sum of ENN A, A1, B, and B1.

**Table 2 toxins-14-00463-t002:** Sensitivity of *F. graminearum* and *F. poae* Strains to Triazole Fungicides.

% Inhibition of Mycelial Growth of *Fusarium* Species								
Fungicides	Caramba (metconazole)		Prosaro (prothioconazole + tebuconazole)		Folicur (tebuconazole)		Proline (prothioconazole)	
Dose (mg ai/L)	0.01	0.1	0.01	0.1	0.01	0.1	0.01	0.1
*Fusarium* strains	Mean	Mean	Mean	Mean	Mean	Mean	Mean	Mean
*F. graminearum*	43 ± 5.1 a*	55 ± 4.6 a	39 ± 2.6 a**	51 ± 3.3 a**	2.5 ± 1.1 a*	38 ± 3.2 a*	28 ± 1.9 a*	42 ± 2.2 a**
*F. poae*	34 ± 3.2 b	59 ± 6.2 a	25 ± 1.9 b	29 ± 1.8 b	14 ± 0.5 b	28 ± 2.1 b	21 ± 1.4 b	29 ± 2.1 b

Note: Mean ± SE refers to combined average radial growth of 15 *F. poae* and 15 *F. graminearum* selected strains. Means of % inhibition followed by the same letter within the same fungicide are not significantly different at *p* = 0.05 (*) and *p* = 0.01 (**) based on Student’s *t*-test.

**Table 3 toxins-14-00463-t003:** Effect of Previous Crop and Year on DON and NIV Concentrations in Barley Grains (2017–2019).

**Fixed Effects**	***p*-Value**			
	**DON**		**NIV**	
Previous Crop (PC)	0.01 *		0.34	
Year (2017–2019)	0.10		0.15	
PC * Year	0.02 *		0.29	
**Previous Crop**	**DON (mg kg^−1^)**		**NIV (mg kg^−1^)**	
	**Mean**	**SE**	**Mean**	**SE**
Canola (*n* = 73)	0.06 b	0.02	0.41 a	0.07
Cereal (*n* = 28)	0.23 a	0.07	0.52 a	0.10
Flax (*n* = 17)	0.14 ab	0.04	0.29 a	0.08

Notes: *n* = number of previous crop fields included in this analysis for Mean ± Standard Error (SE). Means of mycotoxin concentrations followed by the same letter or * within the same toxin are not significantly different at *p* ≤ 0.05.

## Data Availability

Not applicable.

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
