# Peer review of "Implications of Crop Rotation and Fungicide on Fusarium and Mycotoxin Spectra in Manitoba Barley, 2017–2019"

_toxins, 2022, doi:10.3390/toxins14070463_

Round 1
Reviewer 1 Report
The manuscript deals with the occurrence of mycotoxins in barley grains and species composition of fungi Fusarium naturally occurring in barley fields in the period 2017-2019. The manuscript is well written, however, some points have to be corrected. The details are listed below:
L12: nivalenol with lowercase letter
L14-15: indicate concentration of mycotoxins
L17: write active ingredients instead trade names
L97: respectively
L104: do not use italics for disease name
L126-128: transfer to the Fig. 2 caption. ‘crosses and horizontal lines in the boxes’ are not visible
L135: in the Table 1 is 95.5%
L148-150: check ranges of mycotoxins with Table 1. There are some differences
L174-179: also use active ingredients
L180: ‘except for Caramba at 0.1 mg active ingredient L-1’ – and Foliar at 0.01 mg
L204-206: ‘… United States and Canada, while in cooler regions of Europe F. graminearum, F. culmorum and F. avenaceum are predominant’ (see Lozowicka et al. Impact of Diversified Chemical and Biostimulator Protection on Yield, Health Status, Mycotoxin Level, and Economic Profitability in Spring Wheat (Triticum aestivum L.) Cultivation. Agronomy 2022, 12, 258. https://doi.org/10.3390/agronomy12020258)
L246: Fusarium in italics
L350: in this point indicate weather conditions in 2017-2019
L351: indicate range of area of the fields
L356-357: indicate precisely names of preceding crops
L363: indicate species which were analyzed by multiplex PCR
L380-382: indicate full names of mycotoxins. Were the mycotoxins analyzed in replicates?
L418: why only these two species were selected for sensitivity testing, while others were also determined in the barley samples?
L454: triazole
Author Response
Reviewer 1
The manuscript deals with the occurrence of mycotoxins in barley grains and the species composition of fungi Fusarium naturally occurring in barley fields from 2017-2019. The manuscript is well written, however, some points have to be corrected.
Our response:
Thanks for the encouragement. We have made corrections based on the suggestions from the reviewer.
The detailed corrections made:
L12: nivalenol with a lowercase letter
Our response: changed to the lowercase letter at L12
L14-15: indicate the concentration of mycotoxins
Our response: the concentrations of NIV and DON were added in L14 and L15.
L17: write active ingredients instead trade names
Our response: the names of active ingredients were provided in L18
L97: respectively
Our response: spelling mistake was corrected at L97
L104: do not use italics for disease name
Our response: italic for Fusarium was removed at L106
L126-128: transfer to the Fig. 2 caption. ‘crosses and horizontal lines in the boxes’ are not visible
Our response: L126-L128 was moved into the Fig2 caption, and Fig2 was modified to increase the clarity.
L135: in Table 1 is 95.5%
Our response: corrected in L135
L148-150: check ranges of mycotoxins with Table 1. There are some differences
Our response: the ranges of average concentration were double-checked and corrected L147-L150.
L174-179: also, use active ingredients
Our response: changed to the active ingredients at L175.
L180: ‘except for Caramba at 0.1 mg active ingredient L-1’ – and Foliar at 0.01 mg
Our response: good suggestions. The correction was made at L181
L204-206: ‘… United States and Canada, while in cooler regions of Europe F. graminearum, F. culmorum and F. avenaceum are predominant’ (see Lozowicka et al. Impact of Diversified Chemical and Biostimulator Protection on Yield, Health Status, Mycotoxin Level, and Economic Profitability in Spring Wheat (Triticum aestivum L.) Cultivation. Agronomy 2022, 12, 258. https://doi.org/10.3390/agronomy12020258)
Our response: great suggestion. The reference was added at L207-209
L246: Fusarium in italics
Our response: corrected at L246
L350: in this point indicate weather conditions in 2017-2019
Our response: good point. The weather condition was mentioned in L347.
L351: indicate the range of area of the fields
Our response: the latitude and longitude of the areas were added in Ln355-356
L356-357: indicate precisely the names of preceding crops
Our response: the names of preceding crops were added in Ln361
L363: indicate species which were analyzed by multiplex PCR
Our response: species names were added in Ln368
L380-382: indicate full names of mycotoxins. Were the mycotoxins analyzed in replicates?
Our response: the full names of the mycotoxins were added in Ln385 – LN 389. Due to the cost associated with HRMS, we could only test a small selected set of samples in replications (i.e. the ones with high DON and NIV concentrations). The majority of samples were only analyzed once with HRMS.
L418: why only these two species were selected for sensitivity testing while others were also determined in the barley samples?
Our response: In this study, we focused on F. graminearum and F. poae due to their predominance in barley and oat. It has been challenging to obtain enough isolates for the fungicide test due to the low isolation frequency of F. sporotrichioides and F. avenaceum,
L454: triazole
Our response: corrected at L454
Reviewer 2 Report
The manuscript presents the occurrence of Fusarium species and the mycotoxins produced, in correlation with crop rotation and the application of fungicides in barley, in Manitoba-Canada in the years 2017-2019. Identification of fungal species by the PCR method showed a dominance of Fusarium poae, followed by F. graminearum and F. sporotrichioides, and the UHPLC-HRMS method showed the prevalence of the mycotoxins nivalenol and deoxynivalenol. DON contamination in barley was higher in cereal precursor crops, and F. poae was more resistant to triazole fungicides.
The manuscript provides sufficient introductory information, the methods are adequately described, and the results are analyzed in-depth and presented in detail. The conclusions are supported by the results.
I consider that the manuscript meets the conditions for publication in Toxins.
I recommend adding some information before publishing:
1. Mention global areas of barley cultivation.
2. The geographical coordinates of the provinces under study and their type of climate.
3. Clear mention of the anthesis period of barley in Canada (June - July ???).
4. The values ​​of precipitation and air temperature during the anthesis every year, because there are big differences between the averages and the maximums of the mycotoxins (Table 1). In this regard, I note that FAOSTAT shows large differences in barley productivity in Canada in 2017-2019.
FAO. FAOSTAT. Available online: https://www.fao.org/faostat/en/#data/QCL/visualize
5. Mention the maximum limits allowed in Canada for each mycotoxin in barley. 6. Comparative distribution of Fusarium in Canada and Europe.
Backhouse, D. Global distribution of Fusarium graminearum, F. asiaticum and F. boothii from wheat in relation to climate. Eur. J. Plant Pathol. 2014, 139, 161–173.
Pasquali, M.; Beyer, M.; Logrieco, A.; Audenaert, K.; Balmas, V.; Basler, R.; Boutigny, A.-L.; Chrpová, J.; Czembor, E.; Gagkaeva, T.; et al. A European Database of Fusarium graminearum and F. culmorum Trichothecene Genotypes. Front. Microbiol. 2016, 7, 406.
7. Mention the manufacturer for each fungicide.
8. Add DOI of each reference, if available.
Author Response
Reviewer2
The manuscript presents the occurrence of Fusarium species and the mycotoxins produced, in correlation with crop rotation and the application of fungicides in barley, in Manitoba-Canada in 2017-2019. Identification of fungal species by the PCR method showed a dominance of Fusarium poae, followed by F. graminearum and F. sporotrichioides, and the UHPLC-HRMS method showed the prevalence of the mycotoxins nivalenol and deoxynivalenol. DON contamination in barley was higher in cereal precursor crops, and F. poae was more resistant to triazole fungicides.
The manuscript provides sufficient introductory information, the methods are adequately described, and the results are analyzed in-depth and presented in detail. The conclusions are supported by the results.
I consider that the manuscript meets the conditions for publication in Toxins.
Our response:
Thanks for the encouragement. We have made corrections based on the suggestions from the reviewer.
The detailed corrections made:
- Mention global areas of barley cultivation.
Our response: global barley acreage and production were mentioned in L32.
- The geographical coordinates of the provinces under study and their type of climate
Our response: the geographical coordinate of the province was added Ln 355-356. The type of climate in Manitoba is added in Ln 355 to 356.
- Clear mention of the anthesis period of barley in Canada (June - July ???).
Our response: the anthesis period of barley in Manitoba is added in Ln 357-358.
- The values ​​of precipitation and air temperature during the anthesis every year, because there are big differences between the averages and the maximums of the mycotoxins (Table 1). In this regard, I note that FAOSTAT shows large differences in barley productivity in Canada in 2017-2019.
FAO. FAOSTAT. Available online: https://www.fao.org/faostat/en/#data/QCL/visualize
Our response: the weather data of Manitoba 2016-2019 was added in a new supplement table S3.
- Mention the maximum limits allowed in Canada for each mycotoxin in barley.
Our response: the maximum limits of HT-2, T-2 and DAS in Canada are added in LN83-97
- Comparative distribution of Fusariumin Canada and Europe.
Our response: The information on Fusarium in Europe is added in Ln 210-211.
- Mention the manufacturer for each fungicide.
Our response: the manufacturers of each fungicide are added in LN 428 – 429.
- Add DOI of each reference, if available.
Our response: DOIs of references are added.
Reviewer 3 Report
This is an interesting study which investigated the barley grain collected in Manitoba for Fusarium species spectra and naturally occurring mycotoxins. Findings from this research will assist barley producers with improved understanding of FHB (Fusarium head blight) threat levels and optimizing practices for the best management of FHB in barley. The study is all the more important as FHB is one of the most important diseases on barley not just from western Canada but also other major barley producing regions of the world.
The abstract presents very clear the objectives of the study. The introduction is supported by well selected bibliographic data. All bibliographic sources are fairly recent and correctly mentioned in text. However, before acceptance, authors need to address the below comments:
L. 37-38. Given the mycotoxins potential to affect the health of humans and animals, I suggest that you improve the Introduction part by adding more current references that highlight the dangers of these mycotoxins. One reference only seems insufficient.
L. 41, 246, 466, etc. Please write the scientific name Fusarium in italics and check this carefully in entire manuscript.
L. 351-357. Climatic conditions are one of the main factors in the occurrence and development of mycotoxins. Therefore, I suggest you detail the climatic conditions for each five crop districts of Manitoba.
L. 427. I guess ″3 d″ means 3 days. But I would like you to make this clear, to avoid any confusion.
Author Response
Reviewer 3
This is an interesting study which investigated the barley grain collected in Manitoba for Fusarium species spectra and naturally occurring mycotoxins. Findings from this research will assist barley producers with improved understanding of FHB (Fusarium head blight) threat levels and optimizing practices for the best management of FHB in barley. The study is all the more important as FHB is one of the most important diseases on barley not just from western Canada but also other major barley producing regions of the world.
The abstract presents very clear the objectives of the study. The introduction is supported by well selected bibliographic data. All bibliographic sources are fairly recent and correctly mentioned in text. However, before acceptance, authors need to address the below comments:
Our responses:
Thanks for the comments, We have made corrections based on the suggestions from the reviewer.
The detailed corrections made:
1. L37-38. Given the mycotoxins potential to affect the health of humans and animals, I suggest that you improve the Introduction part by adding more current references that highlight the dangers of these mycotoxins. One reference only seems insufficient.
Our response: Additional references are provided in Ln40-44
2. L41, 246, 466, etc. Please write the scientific name Fusariumin italics and check this carefully in entire manuscript.
Our response: Corrected. The scientific name Fusarium is written in the main text and reference list.
3. L351-357. Climatic conditions are one of the main factors in the occurrence and development of mycotoxins. Therefore, I suggest you detail the climatic conditions for each five crop districts of Manitoba.
Our response: the climate conditions for five crop districts of Manitoba are provided in a new supplement Table (Table S3).
4. 427. I guess ″3 d″ means 3 days. But I would like you to make this clear, to avoid any confusion.
Our response: we changed to 3 days to avoid any confusion (Ln 437)
Round 2
Reviewer 1 Report
The Authors have improved the paper according to the comments. However in L 384-386, mean temperature and precipitation in each year in Manitoba should be added.
Author Response
The mean temperature and precipitation in each year in Manitoba are added in Ln 379-380. We also added the mean temperature and precipitation of Manitoba for each year in Table S3.